# Targeted genome editing by lentiviral protein transduction of zinc-finger and TAL-effector nucleases

**Yujia Cai, Rasmus O Bak, Jacob Giehm Mikkelsen\***

Department of Biomedicine, Aarhus University, Aarhus, Denmark

**Abstract** Future therapeutic use of engineered site-directed nucleases, like zinc-finger nucleases (ZFNs) and transcription activator-like effector nucleases (TALENs), relies on safe and effective means of delivering nucleases to cells. In this study, we adapt lentiviral vectors as carriers of designer nuclease proteins, providing efficient targeted gene disruption in vector-treated cell lines and primary cells. By co-packaging pairs of ZFN proteins with donor RNA in 'all-in-one' lentiviral particles, we co-deliver ZFN proteins and the donor template for homology-directed repair leading to targeted DNA insertion and gene correction. Comparative studies of ZFN activity in a predetermined target locus and a known nearby off-target locus demonstrate reduced off-target activity after ZFN protein transduction relative to conventional delivery approaches. Additionally, TALEN proteins are added to the repertoire of custom-designed nucleases that can be delivered by protein transduction. Altogether, our findings generate a new platform for genome engineering based on efficient and potentially safer delivery of programmable nucleases.

## Introduction

The capacity of designed nucleases, like ZFNs and TALENs, to generate DNA double-stranded breaks (DSBs) at desired positions in the genome has created optimism for therapeutic translation of locus-directed genome engineering. ZFNs and TALENs are chimeric nucleases composed of a custom-designed DNA binding domain fused to the DNA-cleavage domain from the FokI endonuclease that upon dimer formation cleaves the DNA. ZFN- and TALEN-induced DSBs trigger genome editing through cellular repair mechanisms involving either error-prone non-homologous end joining (NHEJ) or homologous recombination (HR) with an available DNA donor template. Designer nucleases have broad applications in biological experimentation (*Urnov et al., 2010*; *Bogdanove and Voytas, 2011*) and have been successfully utilized for the production of gene knockout model animals (*Doyon et al., 2008*; *Geurts et al., 2009*; *Tesson et al., 2011*) and in emerging gene therapies (*Perez et al., 2008*; *Li et al., 2011*, *2013*; *Sun et al., 2012*).

The safety of designer nucleases is of major concern in relation to their use in treatment of human diseases. Thus far, ZFNs and TALENs have been administered to cells by transfection or electroporation of nucleic acids, DNA or RNA, encoding a pair of nuclease proteins (*Urnov et al., 2005*; *Miller et al., 2011*; *Carlson et al., 2012*) or by exploiting viral gene vehicles such as integrase-deficient lentiviral vectors (IDLVs) (*Lombardo et al., 2007*), adeno-associated virus-derived vectors (AAV vectors) (*Ellis et al., 2013*), or adenoviral vectors (*Holkers et al., 2013*). Successful administration of ZFN- or TALEN-encoding genes leads to high intracellular levels of nucleases and furthermore imposes a risk of random insertion in the genome, resulting potentially in prolonged nuclease expression and accumulating events of off-target cleavage. Ideally, ZFNs and TALENs are provided in a 'hit-and-run' fashion allowing short-term and dose-controllable nuclease activity without losing the effectiveness of creating locus-directed DSBs. Towards this goal, ZFNs have been fused to destabilizing domains regulated by small

**\*For correspondence:** giehm@
hum-gen.au.dk

**Competing interests:** The authors declare that no competing interests exist.

**Reviewing editor**: Todd Golub, Broad Institute, United States

**eLife digest** Altering the genetic code of a living organism to produce certain desirable outcomes is the goal of genetic engineering. The field builds on a long history of human attempts to alter genetics, from selective breeding of crops and livestock to genetically modified organisms and gene therapies. Researchers routinely use gene editing to create 'knock-out' mice in which a particular gene is turned off: the researchers can learn more about the function of this gene by watching what happens when it is absent.

As gene editing techniques have grown more sophisticated, they have become an increasingly promising tool for treating diseases that are caused by gene mutations. The aim of this work is to replace faulty genes with genes that work properly. However, it has been difficult to adapt genetic engineering techniques so that they can be used safely in humans.

Scientists have created customized enzymes called nucleases that can remove specific genes, but it has been a challenge to get these nucleases into cells in the first place. A virus can be used to deliver the genes that encode these nucleases into the DNA of a cell, but this approach can lead to the production of too many nucleases and to the removal of more genes than intended.

Now Cai et al. have developed a 'hit-and-run' method for getting the nucleases into cells and making them active only for a short period of time. This method involves using a virus to deliver two different nucleases to a cell. Once inside the cell, the viruses released the nucleases, which were able to remove up to one-quarter of their gene targets, with relatively few errors, in the time that they were active.

Next, Cai et al. added gene patches—new genes to replace those removed by the nucleases—to the viruses. This 'cut and patch' strategy was successful in up to 8% of the treated cells. The results also suggest that this approach is safer than other gene-editing techniques.

molecules to attenuate ZFN toxicity (*Pruett-Miller et al., 2009*). Moreover, by exploiting the cell-penetrating capability of ZFNs, targeted gene disruption has recently been achieved by direct cellular delivery of purified ZFN proteins (*Gaj et al., 2012*). Although such approach may require multiple treatments due to the reduced cellular uptake of proteins (*Mellert et al., 2012*), recent findings suggest that ZFN uptake may be further improved by ligand-mediated endocytosis (*Chen et al., 2013*). However, for gene correction by homology-directed repair such strategies would need to be combined with other means of delivering the donor template.

It has been known for decades that retroviruses can tolerate the incorporation of heterologous proteins (*Jones et al., 1990*; *Weldon et al., 1990*). Lentiviral particles (LPs) have been engineered to carry foreign proteins for the purpose of visualizing the intracellular behavior of the virus during infection (*McDonald et al., 2002*; *Jouvenet et al., 2008*) and altering the viral integration profile (*Bushman, 1994*; *Goulaouic and Chow, 1996*; *Bushman and Miller, 1997*), as well as for ferrying antiviral (*Okui et al., 2000*; *Ao et al., 2008*) and antitumor (*Link et al., 2006*; *Miyauchi et al., 2012*) protein therapeutics. As the delicate structural composition of HIV-1-derived lentiviral particles is easily disturbed by an inappropriate load of nonviral proteins, leading to suboptimal vector yields and/or reduced transduction capability, various strategies for transducing heterologous protein cargo have been scrutinized. In early strategies, the accessory HIV-1 protein Vpr was adapted as a carrier of fused proteins (*Wu et al., 1995*). Recently, Vpr fusions have been shown also to ferry Cre recombinase (*Michel et al., 2010*) and I-SceI meganuclease (*Izmiryan et al., 2011*) into transduced cells. However, HIV-1 virions incorporate relatively few copies of Vpr (estimated 700 copies Vpr per virion [*Swanson and Malim, 2008*]), and the therapeutic potential of such approach may be hampered further by the known toxicity of the Vpr protein (*Tachiwana et al., 2006*). Alternatively, nonviral proteins may be packaged in LPs as part of the Gag polypeptide, as was previously shown for reporter proteins like GFP (*Aoki et al., 2011*) and the apoptosis-inducing caspase 3 protein (*Miyauchi et al., 2012*). During virion maturation, Gag is processed by the viral proteins into shorter proteins constituting the structural—and most abundant—proteins of the virus particle. It is estimated that each virion contains 5000 copies of Gag and 250 copies of GagPol (*Swanson and Malim, 2008*). We recently adapted LPs for the delivery of the *piggybac* DNA transposase (*Cai et al., 2014*). The transposase was released from Gag in the virus particles in a protease-dependent manner and found to be able to facilitate efficient DNA transposition in

transduced cells. In yet another strategy, heterologous proteins fused to the integrase in the Pol region of the GagPol polypeptide were successfully delivered by protein transduction (*Schenkwein et al., 2010*).

In this study, we describe the use of lentivirus-derived particles as carriers of designer nucleases for safe administration of ZFN and TALEN proteins fused to lentiviral Gag precursors. We produce ZFN-loaded lentiviral particles that induce high-efficiency gene disruption with a favorable on-target/off-target ratio in safe genomic harbors like the *CCR5* locus. Also, gene disruption and repair is evident in cells treated with particles carrying TALEN proteins. Successful incorporation of nuclease proteins within lentiviral particles allows co-delivery of nucleases and the donor template for homology-directed repair. Our findings demonstrate targeted and programmable gene repair in the human genome by delivery of both 'scissors' and 'patch' in a single combined protein and gene vehicle.

## Results

### Incorporation, processing, and transduction of ZFNs in LPs

To incorporate ZFN proteins in LPs, we fused ZFNs to the N-terminus of Gag containing also an intervening heterologous phospholipase C-δ1 pleckstrin homology (PH) domain thought to promote the recruitment of Gag and GagPol to the membrane (*Urano et al., 2008*). We began by incorporating ZFNs targeting the *egfp* reporter gene (*Urnov et al., 2005*) into the *gag* gene of a packaging construct harboring the IN D64V mutation (*Figure 1A*), rendering the viral integrase (IN) incapable of catalyzing vector insertion. An HIV-1 protease cleavage site (SQNY/PIVQ) was included between the ZFN and the PH domain to allow the release of functional ZFN proteins during particle maturation. LPs harboring HA-tagged left and right *egfp*-targeting ZFNs, respectively, were produced separately and analyzed for their particle content by Western blot analysis (*Figure 1B,C*). For both ZFN-loaded LPs, we detected HA-tagged ZFNs of the predicted, full-length size (~37.5 kDa), demonstrating that the majority of the virion-associated ZFNs was correctly released from the Gag and GagPol polypeptides during particle maturation. The identity of this band was confirmed by control analyses including plasmid-encoded HA-tagged ZFN protein (*Figure 1—figure supplement 1*). Nevertheless, both longer and shorter forms of the ZFNs were evident (indicated by ZFN* in *Figure 1B,C*), suggesting that processing was not complete and that proteolytic cleavage had also occurred outside the inserted cleavage site, despite the fact that neither the ZFNs nor the PH domain contained naive HIV-1 cleavage sites. Release of ZFN and p24 (*Figure 1B,C*, left and right panels, respectively) from ZFN-PH-GagPol polypeptides was inhibited by treatment of the virus-producing cells with the HIV-1 protease inhibitor Saquinavir (SQV), confirming that polypeptide cleavage, and the resulting release of ZFN proteins, was dependent on the HIV-1 protease.

### Efficient gene disruption by ZFN proteins delivered by LPs

Next, we transduced HEK293-eGFPmut reporter cells harboring a mutated *egfp* gene with increasing doses of LPs, designated LP-ZFNLR(gfp), harboring both left and right *egfp*-targeting ZFNs (ZFNL(gfp) and ZFNR(gfp)). ZFN protein transduction caused dose-dependent cleavage of *egfp* as measured by detection of NHEJ-induced sequence alterations using the Surveyor nuclease assay (*Figure 1D*). The presence of indels inside *egfp* was confirmed by sequence analysis of cloned PCR fragments obtained from cells treated with 600 ng p24 LP-ZFNLR(gfp) (*Table 1—source data 1*). Among a total of 42 analyzed clones, 8 were found to contain *egfp* gene disruptions, providing an unbiased disruption frequency of 19% (*Table 1*). Notably, ZFN-directed cleavage and subsequent error-prone repair by NHEJ was detectable as early as 12 hr posttransduction (*Figure 1E*).

To evaluate the versatility of this approach, we produced LPs loaded with ZFNs targeting the endogenous *CCR5* and *AAVS1* loci (designated as LP-ZFNLR(CCR5) and LP-ZFNLR(AAVS1), respectively) and exposed HEK293 cells, normal human dermal fibroblasts (NHDFs), and primary human keratinocytes (HKs) to increasing dosages of these LPs. For both loci and in all cell types, we observed significant levels of disruption and found that such disruption occurred in an LP dose-dependent manner (*Figure 2*). In HEK293 cells and HKs, we consistently observed an extra band (indicated by * in *Figure 2*, left panels) that was confirmed by sequencing to originate from the *CCR5* Δ32 allele present in these cells. To evaluate the extent of cleavage and error-prone repair at the *CCR5* and *AAVS1* loci induced by LP-delivered pairs of ZFNs, we cloned and sequenced PCR products encompassing the targeted regions. For both loci and in all three cell types, numerous different indels could

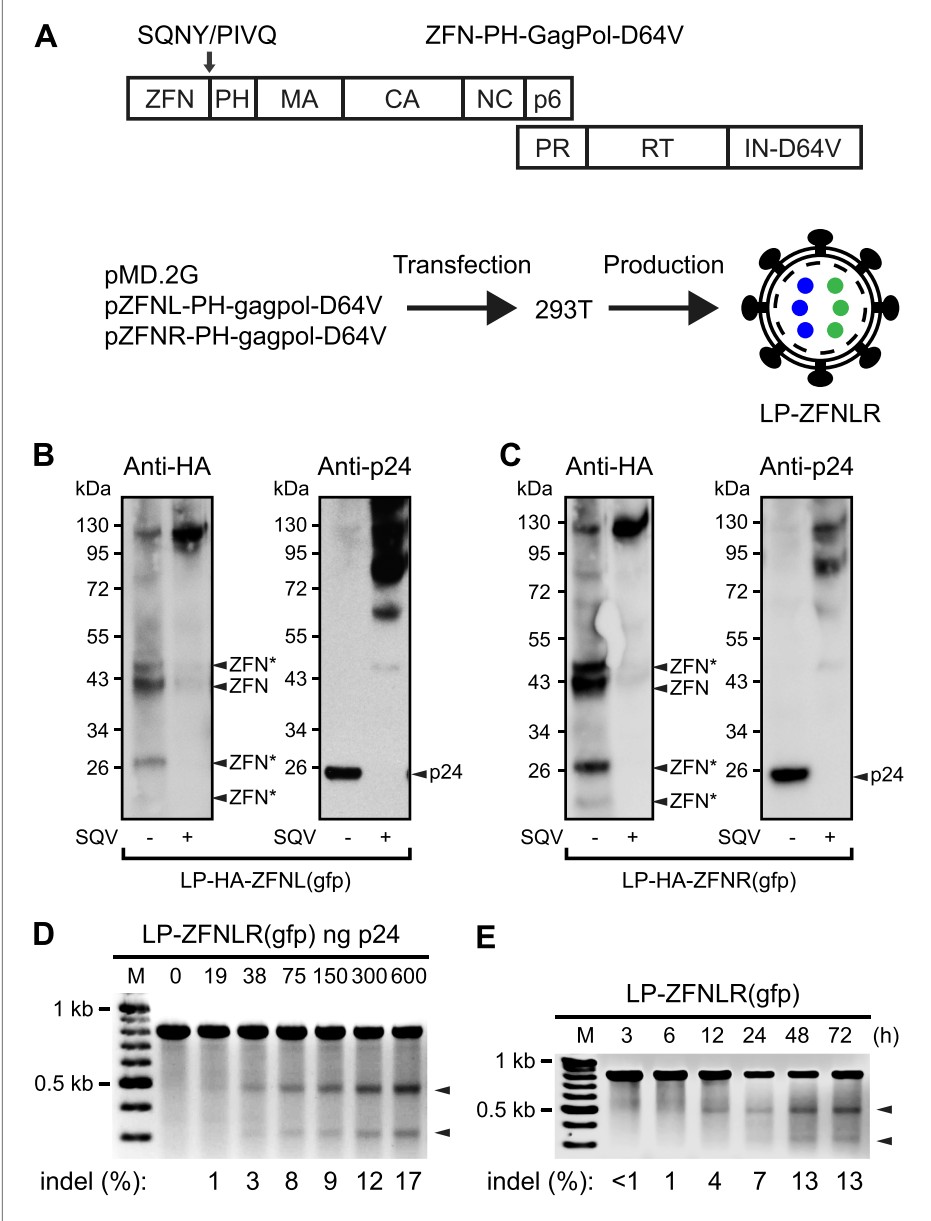

**Figure 1**. Targeted *egfp* gene disruption by lentiviral delivery of ZFN proteins. (**A**) Schematic representation of the composition of Gag and GagPol polypeptides encoded by ZFN-encoding packaging constructs (top) and the production of ZFN-loaded LPs (bottom). Gag is composed of the N-terminal ZFN domain, an HIV-1 cleavage site (SQNY/PIVQ), the phospholipase C-δ1 pleckstrin homology (PH) domain, matrix (MA), capsid (CA), nucleocapsid (NC), and p6, whereas Pol consists of protease (PR), reverse transcriptase (RT), and integrase harboring the D64V mutation (IN-D64V). LPs harboring two types of ZFNs (indicated by blue and green dots inside the virion) are produced by co-transfecting 293T cells with pMD.2G (encoding VSV-G surface protein) and pZFNL-PH-gagpol-D64V and pZFNR-PH-gagpol-D64V encoding ZFNL and ZFNR, respectively. (**B**) and (**C**) Analysis of the contents of LPs by Western blot using HA- and p24-specific antibodies. HA-tagged ZFNs were incorporated in this LP batch, allowing detection of ZFNs and ZFN derivatives (left panel). ZFNs originating from non-intentional processing or cleavage at cryptic HIV-1 cleavage sites are indicated by arrows labeled with ZFN*. The same membrane was stripped and re-used for detection of p24 (right panels). It is indicated below each panel whether 0.2 μM of the protease inhibitor Saquinavir (SQV) was included during LP production. (**D**) *egfp* gene disruption by protein transduction of ZFNs in HEK293-eGFPmut reporter cells as measured by Surveyor nuclease-based detection of DNA mismatches. Cells were harvested for analysis 24 hr posttransduction. Arrowheads point to the specific cleavage products. Quantified locus modification rates (indel %) are indicated below relevant lanes. (**E**) *egfp* gene

*Figure 1. Continued on next page*

*Figure 1. Continued*
disruption by ZFN proteins at different time points after transduction. HEK293-eGFPmut reporter cells were
transduced with 300 ng p24 LP-ZFNLR(gfp). Locus modification rates (indel %) are provided below the gel.
The following figure supplements are available for figure 1:
**Figure supplement 1**. Validation of the release of full-length ZFN protein within ZFN-loaded LPs.

be identified in the targeted region of the genome (*Table 1—source data 2 and 3*). The rates of gene
disruption by both *CCR5*- and *AAVS1*-targeting ZFNs (using an amount of LPs corresponding to 200 ng
p24) are summarized in *Table 1*. For the three cell types, the percentage of sequenced clones harboring
targeted disruptions in the *CCR5* locus ranged from 17% in HEK293 cells and NHDFs to 24% in primary
HKs, whereas from 13% to 20% of the analyzed *AAVS1* loci, depending on the cell type, harbored
indels. In summary, our findings demonstrate potent targeted gene disruption and knockout by lentiviral
delivery of ZFN pairs targeting predetermined loci in the human genome.

## Targeted genome editing by 'all-in-one' lentiviral vectors

We next set out to engineer ZFN-loaded lentiviral vectors with the capacity to not only disrupt but also
edit targeted genes through HR. By packaging a vector genome harboring a HR donor sequence into
LPs, we reasoned that ZFNs and donor could be co-delivered in such 'all-in-one' lentiviral vectors
(*Figure 3A*). We generated, therefore, a donor vector plasmid carrying a 2.1-kb long sequence with
homology to the *egfp* cassette present in the HEK293-eGFPmut cell line. This vector did not itself express

**Table 1.** Summary of the targeting rates obtained by sequence-based identification of locus-targeted
indels after LP-directed delivery of ZFN proteins

| | Cell type | | |
| Target locus | HEK293 | Normal human dermal fibroblasts | Primary human keratinocytes |
| --- | --- | --- | --- |
| *egfp* | 8/42 (19%)* | N/A | N/A |
| *CCR5* | 7/42 (17%) | 8/46 (17%) | 11/45 (24%) |
| *AAVS1* | 6/46 (13%) | 7/43 (16%) | 8/40 (20%) |

Provided ratios indicate the number of alleles with indels out of the total number of analyzed alleles after treatment
with 600 ng p24 LP-ZFNLR(gfp), 200 ng p24 LP-ZFNLR(CCR5) or 200 ng p24 LP-ZFNLR(AAVS1), respectively.
*indicates that data were obtained in HEK293-eGFPmut reporter cells; N/A, not available.
**Source data 1**. Sequences of *egfp* gene disruption by ZFN protein transduction in the HEK293-eGFPmut reporter
cells. Genomic DNA of HEK293-eGFPmut reporter cells transduced with 600 ng p24 LP-ZFNLR(gfp) was used as PCR
template for amplification and subsequent cloning of the part of the *egfp* gene encompassing the region recognized
by the two ZFNs. The wild-type sequence is shown at the top. The net change of length caused by indels is indicated
to the right of each sequence. Green dashes represent deleted nucleotides, red lower case letters represent nucleotide
substitutions, whereas blue lower case letters illustrate inserted nucleotides. If one particular sequence appeared in
more than one clone, the exact number of clones with this particular sequence is provided in parenthesis.
**Source data 2**. Sequences of *CCR5* gene disruption by ZFN protein transduction of HEK293, NHDFs and HKs.
Genomic DNA of cells transduced with 200 ng p24 LP-ZFNLR(CCR5) was used as PCR template for amplification
and subsequent cloning of a *CCR5* amplicon encompassing the region recognized by the two ZFNs. The wild-type
sequence is shown at the top. Types of indels are indicated as described in the legend to *Table 1—source data 1*.
**Source data 3**. Sequences of *AAVS1* gene disruption by ZFN protein transduction in HEK293, NHDFs and HKs.
Genomic DNA of cells transduced with 200 ng p24 LP-ZFNLR(AAVS1) was used as PCR template for amplification
and subsequent cloning of an *AAVS1* amplicon encompassing the region recognized by the two ZFNs. The wild-type
sequence is shown at the top. Types of indels are indicated as described in the legend to *Table 1—source data 1*.
If one particular sequence appeared in more than one clone, the exact number of clones with this sequence is
provided in parenthesis.

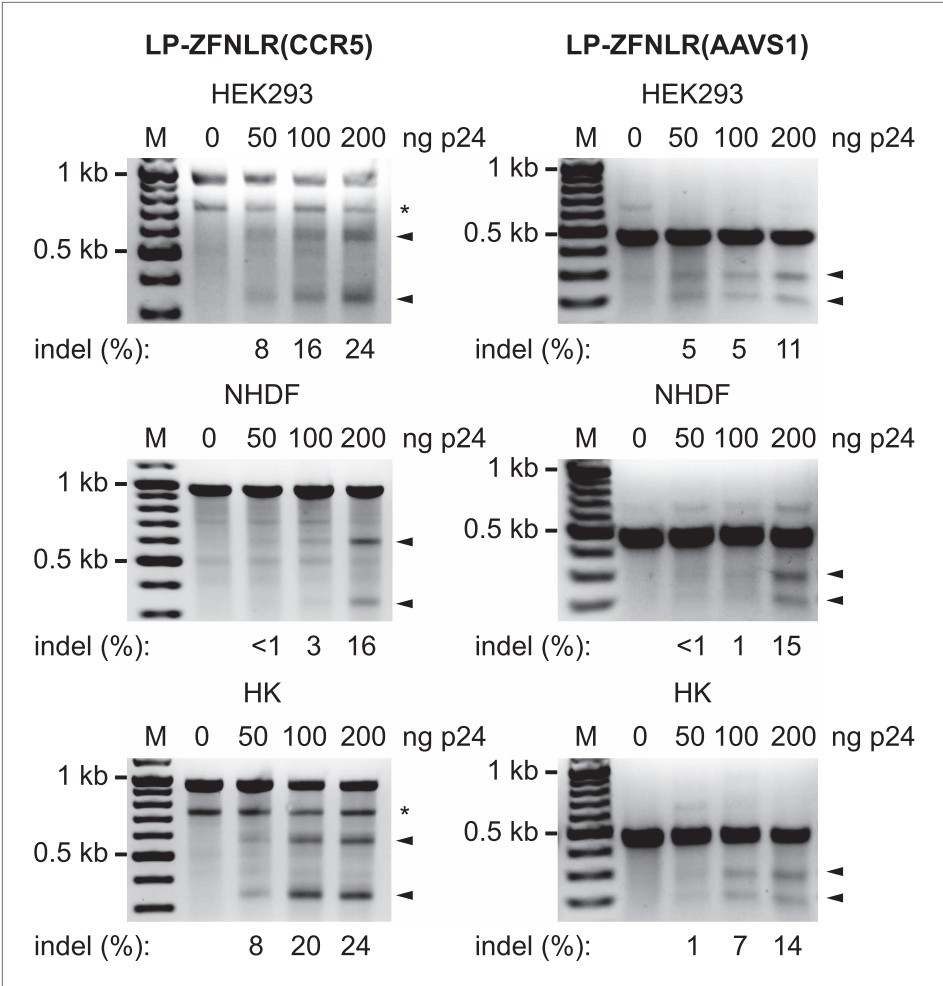

**Figure 2**. Targeted disruption of endogenous genes by protein transduction of ZFNs. HEK293, normal human dermal fibroblasts (NHDFs) and primary human keratinocytes (HKs) were transduced with increasing amounts of LPs containing ZFNs targeting the *CCR5* (left) and *AAVS1* (right) loci. Cells were harvested for analysis 24 hr posttransduction. Arrowheads indicate specific cleavage products, whereas fragments marked with * were generated due to the presence of the CCR5 Δ32 allele in the analyzed cells. Quantified locus modification rates are indicated below relevant lanes.

eGFP, but served as donor for robust repair of *egfp* (leading to eGFP production) in HEK293-eGFPmut cells that were transfected with donor-containing plasmid DNA and treated with increasing dosages of LP-ZFNLR(gfp) (***Figure 3B***). Hence, at a dose of 300 ng p24, LP-ZFNLR(gfp) induced a level of repair that was comparable to that obtained by the efficient co-transfection of plasmid DNA encoding the two ZFNs (***Figure 3B***, compare two right columns). Next, the donor sequence was delivered by integrase-defective lentiviral vectors (IDLV/donor), allowing the reverse-transcribed vector to serve as a recombination donor during HR-directed editing. Co-delivery of IDLV/donor and LP-ZFNLR(gfp) to the HEK293-eGFPmut cells induced gene repair, whereas neither IDLV/donor nor LP-ZFNLR(gfp) alone induced significant levels of eGFP expression (***Figure 3C***).

To investigate whether gene editing could be obtained with viral particles containing both donor RNA and ZFN proteins, HEK293-eGFPmut cells were co-transduced first with two separately produced IDLVs carrying either ZFNL(gfp) or ZFNR(gfp). Both IDLVs carried the vector RNA containing the *egfp* donor sequence. Induced eGFP expression was observed only in cells treated with both IDLVs, indicative of active editing in about 4% of the treated cells, whereas neither of the two vectors alone caused gene editing (***Figure 3D***).

We then created 'all-in-one' IDLVs (IDLV-ZFNLR(gfp)/donor) carrying the *egfp* donor vector as well as both ZFNs, containing thus all the components necessary to facilitate (i) vector transfer, (ii) targeted

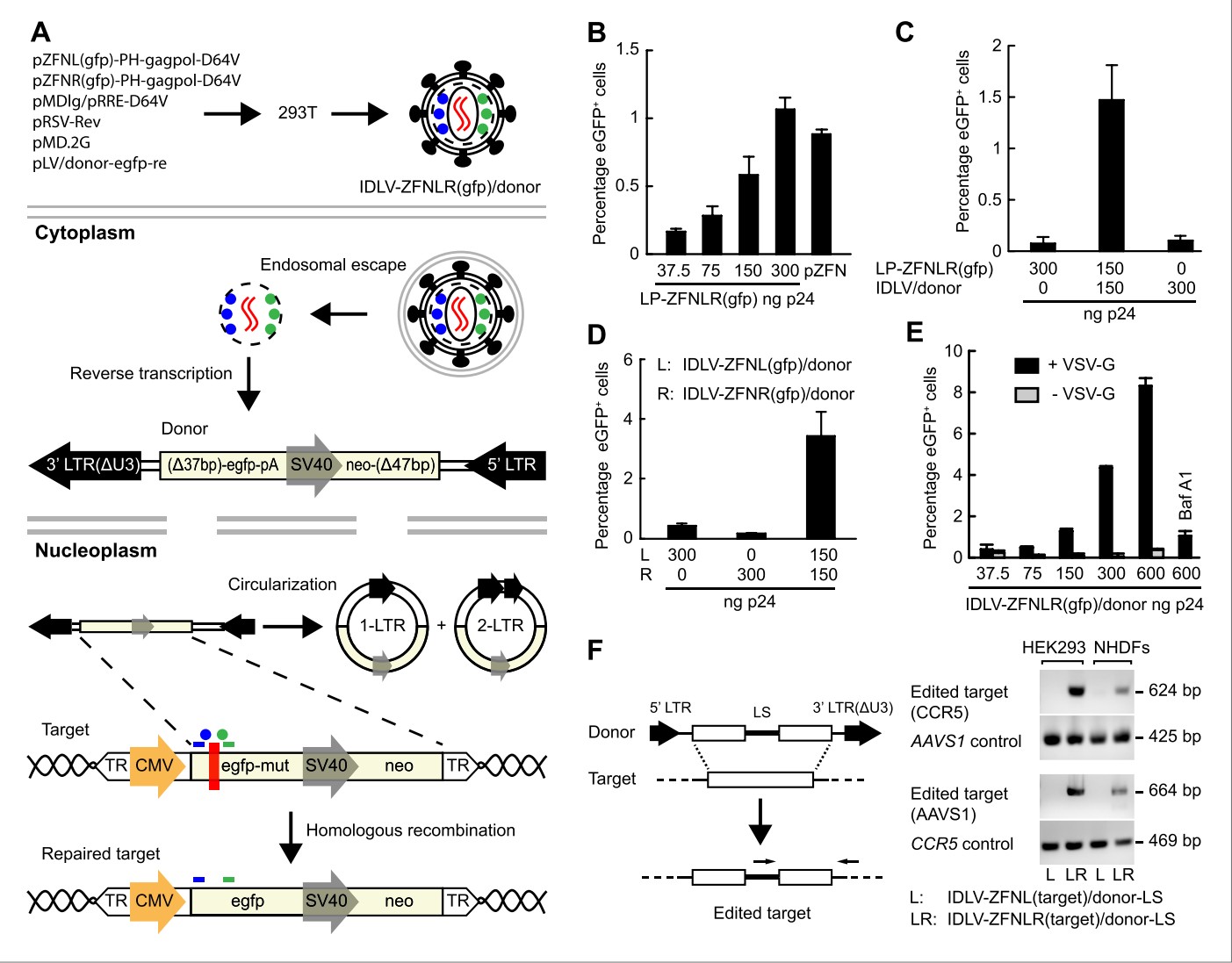

**Figure 3**. Targeted gene editing by 'all-in-one' IDLVs. (**A**) Schematic representation of the production and intracellular action of IDLVs carrying two ZFNs (indicated by blue and green dots) and a vector with the donor sequence for HR-directed repair (indicated in red as an RNA homodimer). Upon endosomal escape and uncoating, the donor sequence is reverse-transcribed to double-stranded DNA that is imported to the nucleus, where it serves as a donor for repair either in the form of linear DNA or as 1-LTR or 2-LTR circles (only HR between linear DNA and the target is shown). Homologous sequences are highlighted in light yellow. The *egfp* gene harboring internal mutations (indicated by a red box) is repaired through ZFN-mediated cleavage and HR using the reverse-transcribed vector as a recombination donor. (**B**–**E**) Correction of the *egfp* gene by lentiviral delivery of ZFN proteins. Flow cytometric analysis was performed 4 days after transduction or transfection. In (**B**), donor plasmid (pLV/egfp-donor-fw) was transfected 6 hr prior to ZFN protein transduction. Co-transfection of donor plasmid and ZFN-encoding plasmid DNA (pZFN) served as a positive control. In (**C**), the donor sequence was provided by IDLV/donor (MOI of 46) co-transduced with LP-ZFNLR(gfp), whereas in (**D**) correction was achieved by co-transduction with two IDLVs (IDLV-ZFNL(gfp)/donor and IDLV-ZFNR(gfp)/donor, respectively), both at an MOI of 9, loaded each with one of the two *egfp*-targeting ZFNs. In (**E**), 'all-in-one' IDLVs (IDLV-ZFNLR(gfp)/donor) induced potent gene correction. Gene editing was measured with virus loads ranging from an MOI of 2 (corresponding to 37 ng p24) to an MOI of 34 (corresponding to 600 ng p24). IDLVs without the VSV-G surface protein as well as reporter cells pretreated with of 1 μM Bafilomycin A1 (Baf A1) served as negative controls. (**F**) Targeted editing at endogenous *CCR5* and *AAVS1* loci. Schematic representation of PCR-based assay used for detection of gene editing at *CCR5* and *AAVS1* loci (left panel). Primers are indicated above the edited target sequence. LS, linker sequence. Gene editing at *CCR5* and *AAVS1* loci in HEK293 cells and NHDFs, as confirmed by PCR (right panel), was obtained with an MOI of 34. PCR fragments amplified from the *AAVS1* locus and the *CCR5* locus served as controls for *CCR5*- and *AAVS1*-directed LS insertion, respectively. The error bars represent ±SD from three independent replicates of the experiment.

The following figure supplements are available for figure 3:

**Figure supplement 1**. Determination of multiplicity of infection (MOI) of IDLVs carrying left and right ZFNs.

formation of DSBs, and (iii) editing by HR. Treatment of the HEK293-eGFPmut cells with this vector resulted in potent induction of eGFP expression in a vector dose-dependent manner, leading to editing at most in more than 8% of the cells treated with ZFN-loaded IDLVs corresponding to 600 ng p24 (*Figure 3E*).

To provide an indication of the number of IDLV particles that was required to obtain such efficient gene correction, we determined the multiplicity of infection (MOI) of donor-containing IDLVs with and without virally incorporated ZFNs (IDLV/donor and IDLV-ZFNLR(gfp)/donor, respectively). Copy numbers of lentiviral DNA and the genomic *albumin* gene in transduced cells were quantified by quantitative PCR (*Figure 3—figure supplement 1A,B*) on total DNA prepared from $1 \times 10^5$ HEK293-eGFPmut reporter cells transduced with IDLVs corresponding to 30 ng p24. Using this approach, we determined MOIs of 9.2 and 1.7 for IDLV/donor and IDLV-ZFNLR(gfp)/donor, respectively (*Figure 3—figure supplement 1C*), demonstrating that the overall transduction capacity of ZFN-loaded IDLVs was reduced more than fivefold relative to standard IDLVs. Hence, the most effective repair rate by IDLV-ZFNLR(gfp)/donor (*Figure 3E*) was obtained with an MOI of approximately 34. It is directly deduced from these analyses that IDLV/donor samples contain $3.1 \times 10^4$ infectious units (IU)/ng p24, whereas preparations of IDLV-ZFNLR(gfp)/donor contain an estimated $5.7 \times 10^3$ IU/ng p24 (*Figure 3—figure supplement 1D*).

Importantly, editing of the *egfp* gene was absent when the particles were not pseudotyped with VSV-G and therefore incapable of transducing the cells (*Figure 3E*). Furthermore, pretreatment of the reporter cells with Bafilomycin A1 (Baf A1), an inhibitor of endosomal escape, markedly reduced the level of editing (*Figure 3E*). We conclude that 'all-in-one' lentiviral vectors containing both ZFN proteins and the donor sequence facilitate high levels of targeted gene repair in a manner that depends on the dose and VSV-G-mediated endocytosis.

To investigate the versatility of such engineered 'all-in-one' lentiviral virions as tools in genomic editing, we generated IDLVs containing pairs of ZFN proteins targeting the *CCR5* and *AAVS1* loci, respectively, and transduced HEK293 cells and NHDFs at an MOI of 34. In these IDLVs, we packaged a vector genome containing a donor sequence with an internal linker sequence (LS) flanked by two homology arms matching either *CCR5* or *AAVS1* (*Figure 3F*, left panel). By PCR amplification using primers recognizing the LS and sequences outside the homology arm, we observed site-directed editing facilitated by HR of both the *CCR5* and *AAVS1* loci in HEK293 cells and NHDFs (*Figure 3F*, right panel). Notably, signs of editing were not evident in cells treated with preparations of IDLVs harboring only one of the ZFNs (ZFNL(CCR5) or ZFNL(AAVS1)), indicating that homology-directed insertion of the linker was achieved only after delivery of both left and right ZFN proteins.

## Targeted gene disruption by lentivirally delivered TALEN proteins

To expand the repertoire of designed nucleases that are compatible with lentiviral protein transduction, we set out next to explore the possibility of delivering TALEN proteins incorporated into lentiviral particles. We designed a shuttle plasmid that is compatible with the Golden Gate assembly method (*Cermak et al., 2011*) and equivalent to pTAL3, allowing us to isolate the DNA segment carrying the assembled TALEN sequence and insert it into GagPol in the correct reading frame (*Figure 4A*). HEK293-eGFPmut reporter cells were transduced with increasing amounts of LP-TALENLR(gfp) carrying both the left and right TALEN proteins designed to recognize sequences flanking the mutations in the *egfp* gene. Gene disruption was identified by Surveyor nuclease-directed detection of mismatches in re-annealed PCR products (*Figure 4B*) and confirmed by sequencing of cloned PCR products. Hence, 3 out of 46 sequenced alleles (6.5%) were found to contain indels as a result of the treatment with LP-delivered TALEN proteins (*Figure 4C*). Moreover, by transfecting the cells with plasmid DNA containing the donor sequence prior to treatment of the cells with LP-TALENLR(gfp), we found that 0.2% of the cells were eGFP positive (as measured 15 days after transduction), whereas control treatments without LPs and donor DNA, respectively did not trigger eGFP expression (*Figure 4D*). These findings demonstrated the capacity of lentivirally delivered TALEN proteins to facilitate targeted gene repair. Notably, Western blot analysis of viral particles harboring an HA-tagged version of one of the two TALEN proteins fused to the GagPol polypeptide unveiled a cleavage pattern indicative of substantial proteolytic cleavage within the TALEN domain of the fusion protein, leading to truncated versions of the protein (indicated by TALEN* in *Figure 4E*). Although a low level of full-length TALEN protein with the same mobility as HA-tagged TALEN protein expressed in cells (*Figure 4E*, upper band marked by 'TALEN') was indeed evident in the particles, these data indicated the presence of cryptic HIV-1 protease cleavage sites internally in the TALEN protein. This suggests that TALEN-directed gene disruption and repair may be further optimized by localizing and eliminating internal protease cleavage sites.

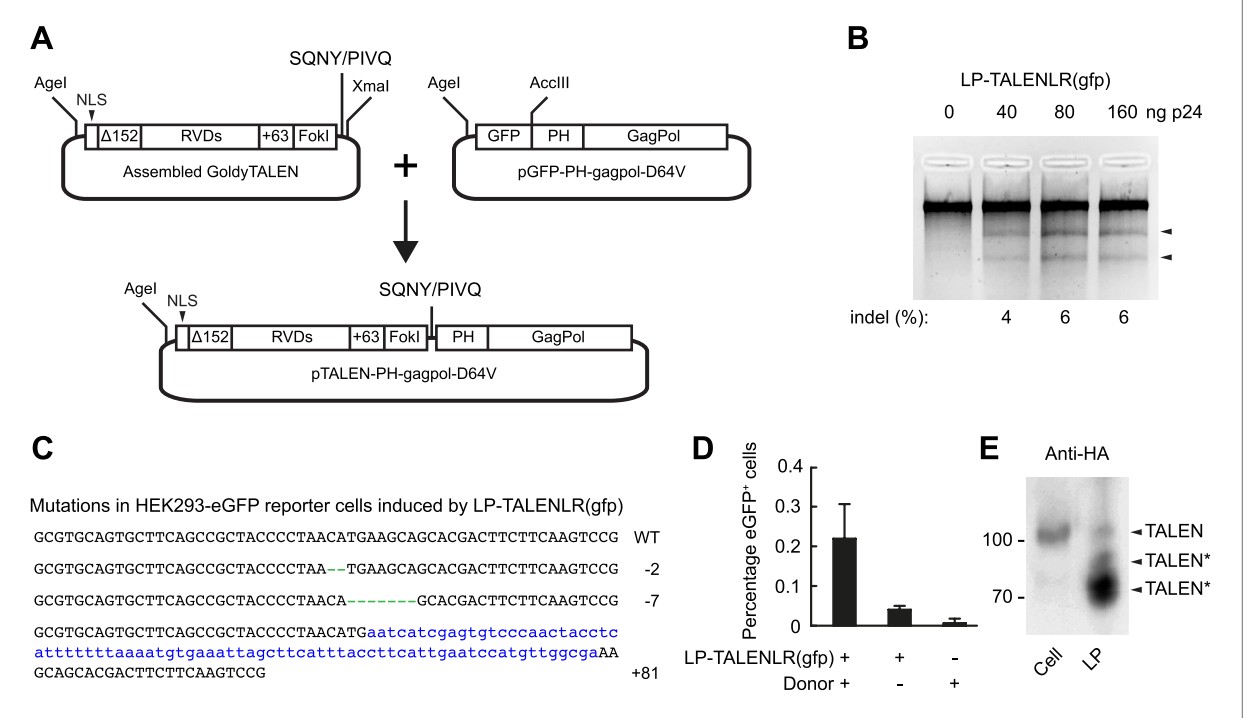

**Figure 4**. Targeted *egfp* gene editing by lentiviral delivery of TALEN proteins. (**A**) Schematic representation of the construction of the TALEN-GagPol polypeptide expression construct. GoldyTALEN was assembled as the Golden Gate assembly method into a shuttle plasmid pC-Goldy-TALEN-PH and was then cut out and cloned into pGFP-PH-gagpol-D64V to get the destination construct pTALEN-PH-gagpol-D64V expressing polypeptides composed of GoldyTALEN and GagPol connected by the HIV-1 protease cleavage site SQNY/PIVQ. NLS, SV40 nuclear localization signal, Δ152, 152 amino acids deletion from the wild-type TALE protein, RVDs, repeat variable di-residues, +63, 63 amino acids following the last repeat, PH, phospholipase C-δ1 pleckstrin homology domain. (**B**) *egfp* gene disruption by protein transduction of TALENs in HEK293-eGFPmut reporter cells as measured by Surveyor nuclease assay 24 hr posttransduction. (**C**) Sequences of *egfp* gene disruption by TALEN protein transduction in the HEK293-eGFPmut reporter cell line. Genomic DNA of HEK293-eGFPmut reporter cells transduced with 160 ng p24 LP-TALENLR(gfp) was used as PCR template for amplification and subsequent cloning of the part of the *egfp* gene encompassing the region recognized by the two TALENs. The wild-type sequence is shown at the top. The net change of length caused by the indels is indicated to the right of each sequence. Green dashes represent deleted nucleotides, whereas blue lower case letters illustrate inserted nucleotides. Three alleles out of 46 sequenced clones were found to be disrupted (disruption frequency: 6.5%). (**D**) Targeted *egfp* gene repair in HEK293-eGFPmut reporter cells. Cells were transfected with 1.8 μg donor plasmid (pLV/egfp-donor-fw) and transduced with 160 ng p24 of LP-TALENLR(gfp) 6 hr later. Cells treated only with LP-TALENLR(gfp) or donor served as negative controls. eGFP expression was analyzed by flow cytometry 15 days posttransduction. (**E**) Analysis of the contents of LP-HA-TALENR(gfp) by Western blot using HA-specific antibody. The left lane shows protein derived from 293T cells expressing right HA-TALEN(gfp) from transfected plasmid pcDNA3.1-Goldy-HA-TALENR(gfp). The expected size of the full-length TALEN is indicated, and truncated TALEN derivatives originating from non-intentional cleavage at cryptic, internal HIV-1 cleavage sites are indicated by an asterisk (*). The error bars represent ±SD from three independent replicates.

## Improved on- vs off-target activity of LP-delivered ZFNs

Nuclease protein transduction facilitates both coordinated delivery of the two proteins and a short boost of activity in transduced cells. As other delivery techniques based on intracellular nuclease production may lead to prolonged nuclease activity, we hypothesized that ZFNs delivered in LPs would have low off-target relative to on-target activity. To test this notion, we compared the activity of *CCR5*-directed ZFNs in the neighboring *CCR2* locus after ZFN delivery into HEK293 cells by protein transduction and plasmid DNA transfection, respectively. The *CCR2* locus has high homology to *CCR5* and has been identified as a major off-target site of *CCR5*-targeting ZFNs (*Perez et al., 2008*; *Lee et al., 2010*). For the *CCR5*-targeting ZFNs employed in this study, only a single nucleotide differs between the *CCR5* and *CCR2* recognition sites of each ZFN. In initial examinations of the potential off-target activity at the *CCR2* locus, we designed a PCR method that would detect NHEJ between fragments created by ZFN activity at both the *CCR5* and *CCR2* locus leading to deletion of the intervening sequence (*Figure 5A*). Indeed, we observed off-target cleavage at the *CCR2* locus by both protein transduction and plasmid transfection but not in the non-treated cells (*Figure 5B*), indicating that both LP-delivered and DNA-encoded ZFNs induced simultaneous double-stranded DNA breaks within the *CCR2* and *CCR5* loci.

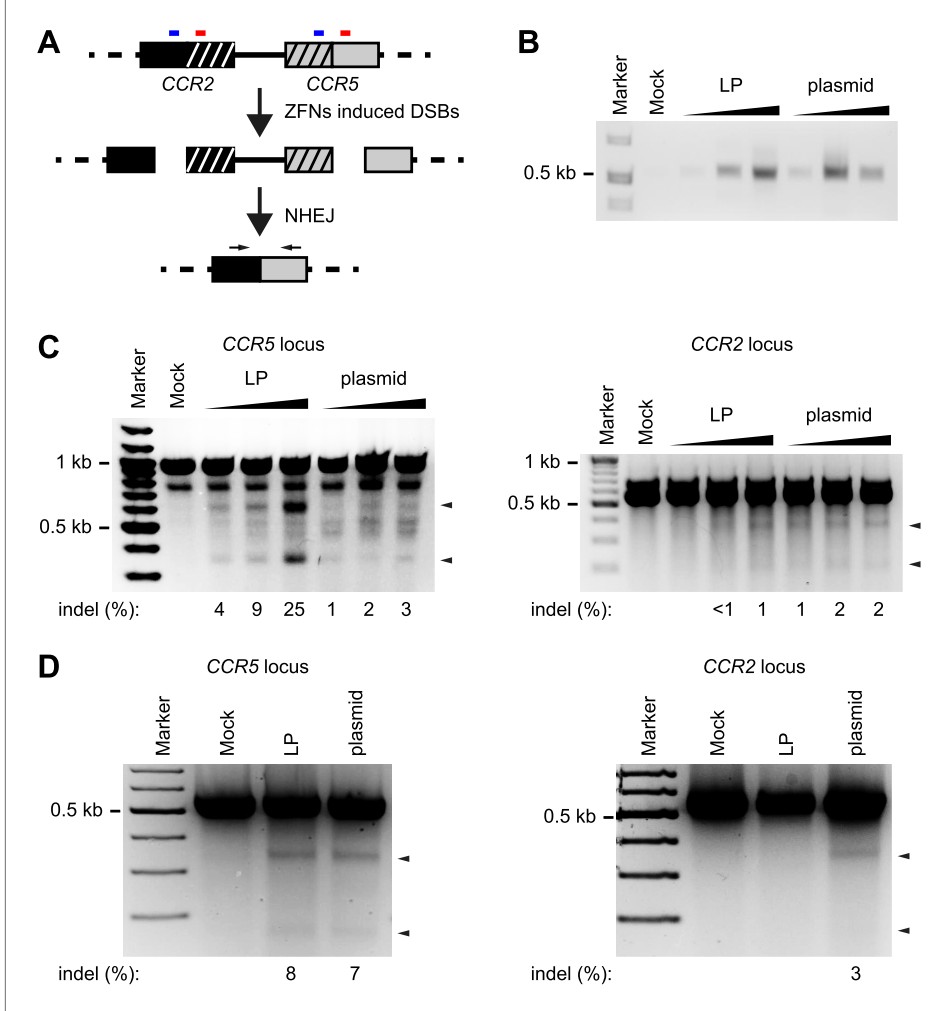

**Figure 5**. On- and off-target activity by *CCR5*-targeting ZFNs delivered by LP protein transduction or plasmid transfection. (**A**) Schematic representation of the PCR-based assay used for detection of simultaneous on-target and off-target cleavage at the *CCR5* and *CCR2* loci, respectively. Small blue and red boxes above each locus indicate the location of the two ZFN recognition sites in the *CCR5* locus and the highly homologous sequences located within the *CCR2* locus. Horizontal arrows indicate the location of sequences recognized by the primers used. (**B**) Detection of off-target cleavage by PCR-based detection of *CCR2-CCR5* fusion fragments. HEK293 cells were seeded at a density of $1 \times 10^5$ cells/well on day 1 and were on day 2 either transduced with increasing amounts of LP-ZFNLR(CCR5) (50, 100, 200 ng p24, respectively; indicated by 'LP') or co-transfected with increasing amounts of the two ZFN-encoding plasmids (50 ng + 50 ng, 100 ng + 100 ng, 200 ng + 200 ng, respectively; indicated by 'plasmid'). On day 3, cells were harvested for genomic DNA purification. (**C**) Disruption within the *CCR5* and *CCR2* loci after LP- and plasmid-directed ZFN delivery. HEK293 cells were treated with increasing amounts of LP-ZFNLR(CCR5) and ZFN-encoding plasmids as in (**B**). Sequence changes introduced by NHEJ-directed repair were identified by Surveyor nuclease-directed detection of mismatches in re-annealed PCR products. Analyses of *CCR5*- and *CCR2*-directed cleavage were performed on the same genomic DNA samples. Arrowheads indicate specific cleavage products. Quantified locus modification rates are indicated below each gel. (**D**) Distinct levels of *CCR2* gene disruption using experimental conditions that support similar levels of *CCR5* disruption after LP- and plasmid-directed ZFN delivery. HEK293 cells were treated by LP transduction (80 ng p24) or plasmid transfection (300 ng + 300 ng), essentially as described in (**B**), facilitating similar levels of gene disruption in the *CCR5* locus. Surveyor nuclease assays were performed as in (**C**) except for the use of an alternative *CCR5* primer set. Quantified locus modification rates are indicated below relevant lanes.

Next, we compared the on-target activity in the *CCR5* locus achieved by the two delivery methods. As shown in *Figure 5C* (left panel) by Surveyor nuclease-directed detection of mismatches in re-annealed PCR products, protein transduction by lentiviral particles induced under the given conditions significantly

higher rates of on-target gene disruption than plasmid transfection. These findings also confirmed the dose-dependent activity of LP-delivered ZFNs. Similarly, when we analyzed for mismatches at the *CCR2* locus, an increasing dose of ZFN-loaded LPs was seen to trigger increased off-target activity (*Figure 5C*, right panel). Importantly, however, a similar level of off-target activity was observed for all doses of transfected plasmid DNA, indicating that LP-delivered ZFNs even at levels that induced markedly higher levels of on-target disruption did not induce increased levels of off-target activity at the *CCR2* locus. These findings showed a reduced relative off-target activity of lentivirus-delivered ZFNs.

To simplify and strengthen the comparison between LP-delivered and plasmid-encoded ZFNs, we established experimental conditions under which the efficiency of on-target gene disruption by LP transduction and plasmid transfection was similar (*Figure 5D*, left panel). By subsequent analyses of off-target disruption within the *CCR2* locus of the same cells, we detected disruption (3%) induced by plasmid-encoded ZFNs, whereas indels and mismatches were not evident after ZFN protein transduction. In summary, our findings lend strong support to the notion that locus targeting by LP-directed protein transduction is both effective and favored by an improved on-target/off-target cleavage ratio.

## Discussion

Engineered nucleases have become driving forces in genomic engineering (*Gaj et al., 2013*). To improve the safe use of such nucleases, it is crucial to scrutinize novel delivery techniques. Lentiviral virions have previously been engineered to incorporate foreign proteins to study different aspects of HIV-1 biology and antiviral strategies (*Wu et al., 1995*; *Fletcher et al., 1997*; *Hermida-Matsumoto and Resh , 2000*; *Hubner et al., 2007*), demonstrating the capability of ferrying protein effectors into target cells. Indeed, delivery of Cre recombinase and I-SceI meganuclease has been achieved by fusing the heterologous protein-of-interest to the HIV-1 accessory protein Vpr (*Michel et al., 2010*; *Izmiryan et al., 2011*). However, none of these enzymes are programmable and therefore can only edit a single artificial target, which would need to be introduced in the target cells prior to editing. Also, as both Cre and I-SceI function as monomers they may be easily adaptable for alternative delivery strategies. In consideration of the increasing focus on engineered nucleases, it is crucial to investigate alternative nuclease delivery strategies.

With the goal of effectively co-delivering two different nuclease proteins and the donor for repair, we turned to a strategy based on transferring the heterologous protein as part of the lentiviral Gag polypeptide. Derivatives of Gag polypeptides are the most abundant proteins in HIV-1-derived parti-cles, and each virion has been estimated to contain approximately 5000 copies of Gag-derived proteins like p24 (*Swanson and Malim, 2008*). Hence, such strategy for transduction of proteins is superior in terms of cargo load. Though IDLVs have been exploited to deliver ZFN-encoding genes (*Lombardo et al., 2007*), there are some clear inherent limitations to this approach. Firstly, random integration of the vector can lead to insertional mutagenesis and sustained expression of ZFNs may cause cytotoxicity or chromosomal aberrations. Secondly, two separate IDLVs are necessary to provide the two ZFNs and a third IDLV needs to be included if a donor template is required for homologous recombination. Single IDLVs encoding both ZFNs have recently been described, but these do not perform as well as two separate IDLVs (*Joglekar et al., 2013*). Thirdly, due to the highly repetitive sequences in TALENs there are major difficulties in delivering these by IDLVs due to an inherent propensity of lentiviral vectors to undergo recombination during reverse transcription (*Mikkelsen and Pedersen, 2000*; *Holkers et al., 2013*).

In this study, we have shown the feasibility of exploiting lentivirus-derived particles as vehicles for different sets of designer nuclease proteins that are released from the Gag polypeptide upon virion maturation. This approach facilitates a short boost of protein activity that drives immediate targeted gene disruption (within 12 hr) in up to near one-fourth of the target alleles in a population of cells without the risks of permanently inserting copies of ZFN- or TALEN-encoding genes into the genome. Moreover, as an alternative to protocols based on transfection of in vitro-transcribed RNA (*Meng et al., 2008*) or the cell-penetrating capability of purified ZFNs (*Gaj et al., 2012*), a lentiviral protein delivery approach benefits from effective lentiviral transduction and can easily be further customized by adapting different lentiviral pseudotypes to direct protein transduction to specific cell types of interest.

The ability to edit predetermined genomic loci, rather than disrupt loci, may be crucial for future therapeutic applications of designer nucleases. We have shown that ZFN proteins (left and right) and HR donor sequences can be incorporated into a single preparation of IDLVs and trigger effective gene repair in transduced cells. At an MOI of approximately 34, such 'all-in-one' vectors edited the reporter

gene in up to 8% of treated cells. We also provide proof-of-principle that 'all-in-one' vectors can target the insertion of additional genetic sequences in the form of a linker sequence, suggesting that HR-mediated insertion facilitated by transduced ZFN proteins has a potential as a tool for targeted lentiviral insertion.

The level of gene repair obtained with IDLVs carrying both ZFNs and the donor template is higher than previously reported repair rates obtained with the same pair of ZFNs under similar conditions using an analogous reporter system (*Urnov et al., 2005*). Although a direct comparison is not possible, we reason that a high rate of repair after protein transduction reflects the combined capacity of LPs to (i) coordinately deliver pairs of nucleases by protein transduction and (ii) provide a donor for homologous recombination which is associated with the pre-integration complex and therefore is actively and efficiently translocated to the nucleus (*Coluccio et al., 2013*). We have previously observed that the uptake of LP-delivered heterologous proteins is extremely limited relative to the very high levels of nuclear protein that can be achieved by standard plasmid DNA transfection (*Cai et al., 2014*). Based on these findings, we currently favor a model by which editing is further supported by the co-localization of ZFN proteins and the donor template within the pre-integration complex. In several aspects, such model mimics the ability of conventional lentiviral vectors to carry the integrase protein along with the reverse-transcribed vector DNA into the nucleus.

In comparison to ZFNs, TALENs have higher modularity and are becoming more routinely used due to the ease of production. However, TALENs are relatively large proteins (~100 kDa), and this may challenge their delivery. In addition, lentiviral delivery of TALEN-encoding genes is further complicated by frequent recombination between the highly similar sequence repeats encoding the DNA-binding domain (*Holkers et al., 2013*), and lentiviral delivery of genes encoding re-coded TALE variants has been reported so far only for transcriptional activation (*Yang et al., 2013*). In this study, we show that TALENs can be delivered as proteins by LPs, facilitating gene disruption or, in the presence of a donor template, repair in a reporter cell line. However, the majority of virally incorporated TALEN subunits are cleaved internally by the HIV-1 protease, which is suspected to limit the overall efficiency. Further work is necessary to identify and eliminate the cryptic viral protease cleavage site to improve the delivery of virus-incorporated TALEN proteins.

It is likely that efficient gene targeting can be achieved in a narrow window of ZFN exposure and that activity outside this time window can cause toxicity without improving gene targeting efficiencies (*Porteus and Baltimore, 2003*; *Pruett-Miller et al., 2009*). To analyze this aspect, we compared ZFN-directed disruption in the *CCR5* target locus and the nearby and highly similar *CCR2* locus. Interestingly, LPs loaded with *CCR5*-directed ZFNs caused disruptions in the *CCR5* locus more effectively than ZFNs expressed from transfected plasmid DNA. In the same cells, disruptions in the off-target *CCR2* locus were evident with all concentrations of plasmid but detectable only with the highest and most effective dose of LPs. Importantly, under experimental conditions that allowed similar rates of disruption using LP-directed ZFN delivery and plasmid-encoded ZFN production, off-target activity within the *CCR2* locus was detected only after ZFN-encoding plasmid delivery. These findings support the notion that LP-delivered ZFNs target a safe harbor in the human genome with an improved on-target/off-target cleavage ratio. We reason that such improvement is a combined effect of efficient transduction and the short-term exposure to transduced ZFN proteins.

With ZFNs and TALENs as early driving forces, designer nucleases stand out as key tools in genomic editing with implications for the development of future gene repair therapies. We anticipate that lentiviral protein transduction of nuclease proteins represents a versatile alternative to current nuclease delivery techniques and will support continued efforts to promote safe genome editing therapies. Uniquely, this approach allows the delivery of the tools for targeted gene editing in a single combined gene and protein vehicle. Likely, genomic engineers will strive to adapt LPs as carriers of Cas9 proteins for RNA-guided genome editing.

## Materials and methods

### Production of lentiviral particles (LPs) and integrase-defective lentiviral vectors (IDLVs)

Throughout this work, the annotation 'LP-ZFNLR(target)' was used to designate lentiviral particles that contain left and right ZFN proteins but do not carry a lentiviral vector genome. Same nomenclature applies to TALEN. A similar type of particle harboring, for example, only the left ZFN was accordingly designated LP-ZFNL(target). When a lentiviral genome was included for example as a carrier of a HR donor sequence, we referred to this vector as an IDLV (due to presence of the IN D64V mutation,

rendering the vector integrase-defective). These vectors were named with the specification of the incorporated ZFNs and the donor that were included. IDLV-ZFNLR(gfp)/donor, for example, contains both *egfp*-directed ZFNs (left and right) and RNAs of which reverse transcription products carry a donor sequence with homology to *egfp*. Of note, we have consistently observed that virions harboring Gag fused N-terminally to foreign proteins are strongly reduced in their capability of transferring vector RNA (most likely due to complications during reverse transcription), but that this capacity can be restored by including unfused GagPol (encoded by pMDlg/pRRE-D64V) in the virions. Preparations of LPs and IDLVs were produced as follows. On day 1, 293T cells were plated at a density of $6 \times 10^4/cm^2$. On day 2, cells were transfected with calcium phosphate precipitates of plasmid DNA. To produce LPs, 293T cells in 15-cm dishes were transfected with 10 µg pMD.2 G, 30 µg pZFNL(target)-PH-gagpol-D64V (targeting either *gfp*, *CCR5*, or *AAVS1*) and 30 µg pZFNR(target)-PH-gagpol-D64V (targeting either *gfp*, *CCR5*, or *AAVS1*). LPs containing TALENs were produced accordingly except using pTALENL(gfp)-PH-gagpol-D64V and pTALENR(gfp)-PH-gagpol-D64V instead of corresponding ZFN constructs. To produce IDLVs that did not incorporate foreign protein, cells in 15-cm dishes were transfected with 9.07 µg pMD.2G, 7.26 µg pRSV-Rev, 31.46 µg pMDlg/pRRE-D64V, and 31.46 µg pLV/egfp-donor-re. To produce IDLVs incorporating ZFN proteins, 293T cells plated in 15-cm dishes were transfected with 9.07 µg pMD.2G, 7.26 µg pRSV-Rev, 15.73 µg pMDlg/pRRE-D64V, 7.8 µg pZFNL(target)-PH-gagpol-D64V (targeting either *gfp*, *CCR5*, or *AAVS1*), 7.8 µg pZFNR(target)-PH-gagpol-D64V (targeting either *gfp*, *CCR5* or *AAVS1*), and 31.46 µg donor-containing vector (either pLV/egfp-donor-re, pLV/CCR5-donor-LS, or pLV/AAVS1-donor-LS). After transfection, the medium was refreshed on day 3, and supernatants were harvested on day 4 and day 5, passed through a 0.45-µm filter (Millipore, Billerica, Massachusetts), and ultracentrifuged at RPM 25,000 at 4°C for 2 hr. Pellets were re-suspended in PBS and stored at −80°C. Concentrations of HIV-1 p24 were measured by ELISA (Zeptometrix, Buffalo, New York) according to the manufacturer's protocol.

## Plasmid construction

pT2/CMV-egfp-mut.SV40-neo containing mutated *egfp* was constructed by inserting a *BsrGI*/*SacII*-digested overlap PCR product into pT2/CMV-egfp(s)-SV40.neo vector (*Staunstrup et al., 2012*). The overlap PCR product was generated by amplifying first two fragments from pT2/CMV-egfp(s).SV40-neo (using primer sets 4684-YJ001R and YJ002F-BGHpA) and fusing these by overlap PCR with primers 4684 and BGHpA (primers are listed in *Supplementary file 1*). The resulting *egfp* cassette harbored both a stop mutation and a frameshift mutation in the sequence flanked by the two ZFN recognition sequences. The lentiviral vector plasmid harboring the *egfp* donor sequence in the forward orientation (pLV/egfp-donor-fw) was created by inserting a PCR product amplified from pT2/CMV-egfp(s).SV40-neo with primers YJ003F and YJ004R into *ApaI*/*XhoI*-digested pLV/RSV-SB100X (*Moldt et al., 2011*). By this procedure, the RSV-SB100X cassette was completely removed and exchanged with the donor sequence. A lentiviral plasmid vector with the *egfp* donor in the reverse orientation (pLV/egfp-donor-re) was created by amplifying the donor sequence from pLV/egfp-donor-fw with the primer set YJ150F-YJ151R and inserting the donor sequence into *ApaI*/*XhoI*-digested pLV/egfp-donor-fw. Each resulting donor vector contained a truncated *egfp* gene (with an upstream 37-bp deletion from the start codon), the SV40 promoter, and a shortened *neo* gene (with a 46-bp deletion as counted from the third position of the stop codon) together constituting a 2100-bp cassette with homology to its target sequence within genomically inserted pT2/CMV-egfp-mut.SV40-neo.

Sequences encoding the zinc-finger (ZF) of known ZFNs targeting the *CCR5* and *AAVS1* loci were synthesized by GenScript according to sequences that have been provided in the literature (*Lombardo et al., 2007*; *Hockemeyer et al., 2009*). To introduce these sequences into a ZFN context, *AflII*/*BamHI*-fragments containing the ZF sequence were inserted into *AflII*/*BamHI*-digested pcDNA3.1-ZFNL(gfp) (*Urnov et al., 2005*) which was kindly provided by Michael C Holmes (Sangamo Biosciences, Richmond, California). The resulting plasmids, in which the ZF sequences were fused to the FokI nuclease domain, were designated pcDNA3.1-ZFNL(CCR5), and pcDNA3.1-ZFNR(CCR5), pcDNA3.1-ZFNL(AAVS1), and pcDNA3.1-ZFNR(AAVS1). We subsequently fused the ZFN-coding sequences to the 5'-end of *gag* gene in pGFP-PH-gagpol, which was kindly provided by Jun Komano, National Institute of Infectious Diseases, Tokyo, Japan. Prior to insertion of the ZFN sequences, the D64V mutation was introduced in the HIV-1 integrase sequence in pGFP-PH-gagpol (creating pGFP-PH-gagpol-D64V) to abolish the conventional lentiviral integration capability. Sequences encoding the left and right gfp-targeted ZFNs were amplified from pcDNA3.1-ZFNL(gfp) and pcDNA3.1-ZFNR(gfp), respectively, with primers YJ112F and

YJ113R. Sequences encoding HA-tagged versions of these ZFNs, HA-ZFNL(gfp) and HA-ZFNR(gfp), were amplified from pcDNA3.1-ZFNL(gfp) and pcDNA3.1-ZFNR(gfp) with primers YJ168F and YJ113R. ZFNL(CCR5)- and ZFNR(CCR5)-encoding sequences were amplified from pcDNA3.1-ZFNL(CCR5) and pcDNA3.1-ZFNR(CCR5), respectively, with primers YJ177F and YJ113R. ZFNL(AAVS1)- and ZFNR(AAVS1)-encoding sequences were amplified from pcDNA3.1-ZFNL(AAVS1) and pcDNA3.1-ZFNR(AAVS1) with the primer sets YJ175F-YJ113R and YJ176F-YJ113R, respectively. The resulting PCR products were digested with *Xma*I and cloned into *Age*I/*Acc*III-digested pGFP-PH-gagpol-D64V to create pZFNL(gfp)-PH-gagpol-D64V, pZFNR(gfp)-PH-gagpol-D64V, pHA-ZFNL(gfp)-PH-gagpol-D64V, pHA-ZFNR(gfp)-PH-gagpol-D64V, pZFNL(CCR5)-PH-gagpol-D64V, pZFNR(CCR5)-PH-gagpol-D64V, pZFNL(AAVS1)-PH-gagpol-D64V, and pZFNR(AAVS1)-PH-gagpol-D64V. To construct pcDNA3.1-HA-ZFNL(gfp) and pcDNA3.1-HA-ZFNR(gfp) for cellular production of HA-tagged ZFNs, HA-ZFN fragments were amplified from pHA-ZFNL(gfp)-PH-gagpol-D64V and pHA-ZFNR(gfp)-PH-gagpol-D64V, respectively, using primer pair YJ211F-YJ212R and inserted into *Afl*II/*Xho*I-digested pcDNA3.1-ZFNL(gfp).

A lentiviral vector plasmid, pLV/CCR5-donor-LS, with a *CCR5*-targeted donor sequence (consisting of two 550-bp homology arms) and an internal linker sequence (LS) was constructed by PCR amplification of the homology arms from genomic DNA of NHDFs with the primer sets YJ193F-YJ206R and YJ195F-YJ196R. The two arms were fused by overlap PCR using primers YJ193F and YJ196R, generating a PCR product with the internal 24-bp LS sequence. The *Apa*I/*Xho*I-digested PCR product was inserted into *Apa*I/*Xho*I-digested pLV/RSV-SB100X (*Moldt et al., 2011*), allowing substitution of the RSV-SB100X cassette with the *CCR5* donor sequence. pLV/AAVS1-donor-LS containing two 600-bp *AAVS1* homology arms flanking an internal LS was constructed by a similar method by which two PCR products were generated with the primer pairs YJ200F-YJ201R and YJ202F-YJ203R. The *Mlu*I/*Xho*I-digested overlap PCR product (generated with primers YJ200F and YJ203R) was inserted into *Mlu*I/*Xho*I-digested pLV/egfp-donor-fw.

To generate pC-Goldy-TALEN-PH, YJ213F-YJ186R-amplified fragments from pC-GoldyTALEN (*Bedell et al., 2012*) were digested by *Spe*I/*Xma*I and inserted into *Xba*I/*Xma*I digested pC-GoldyTALEN. pC-Goldy-HA-TALEN-PH was generated by the same way except using primer YJ218F instead of YJ213F. The repeats of TAL effector were assembled as described except using pC-Goldy-TALEN-PH or pC-Goldy-HA-TALEN-PH to replace pTAL3. The reading frames of the *gfp*-targeting TALENs were cut out by *Age*I/*Xma*I and inserted into the destination vector pGFP-PH-gagpol-D64V digested with *Age*I/*Acc*III. The resulting constructs, pTALENL(gfp)-PH-gagpol-D64V, pTALENR(gfp)-PH-gagpol-D64V, pHA-TALENL-PH-(gfp)-gagpol-D64V, and pHA-TALENR-PH-(gfp)-gagpol-D64V encode polypeptides containing both the TALEN and lentiviral GagPol connected by an HIV-1 cleavage site SQNY/PIVQ. All these constructs contain 63 amino-acid residues at the C-terminus of the TALEN repeat region. pTALEN-PH-gagpol-D64V and a version with an HA-tagged TALEN, pHA-TALEN-PH-gagpol-D64V, were assembled (*Cermak et al., 2011*) and cloned as shown in *Figure 4A*. The repeat variable di-residue sequences of GoldyTALEN for *egfp* targeting were 'NN HD NG NG HD NI NN HD HD NN HD NG NI HD HD' and 'NN NN HD NN NN NI HD NG NG NN NI NI NN NI NI', respectively.

pYJ63-HA-PH-gagpol-D64V was constructed by inserting fragments that were amplified from pC-Goldy-HA-TALEN-PH with primer pair YJ256F-YJ257R into pGFP-PH-gagpol-D64V. Inserts and vector were digested by *Acc*III and *Age*I/*Acc*III, respectively. pcDNA3.1-Goldy-HA-TALEN was generated by inserting fragments amplified from pYJ63-HA-PH-gagpol-D64V with YJ553F-YJ554R into pcDNA3.1-ZFNL(gfp). Inserts and vector were cut by *Afl*II/*Sal*I and *Afl*II/*Xho*I, respectively. HA-tagged TALEN, pcDNA3.1-Goldy-HA-TALENR(gfp), was assembled as Golden Gate cloning by using pcDNA3.1-Goldy-HA-TALEN in place of pTAL3.

## Detection of gene disruption by Surveyor nuclease assay

To determine gene disruption frequencies, we utilized the mismatch detection assay based on the Surveyor nuclease (Transgenomic, Omaha, Nebraska) according to manufacturer's instructions. Briefly, cells treated with ZFN-loaded LPs were harvested 24 hr after transduction unless specific time points were indicated. Genomic DNA was extracted by saturated NaCl and precipitated by absolute ethanol. *egfp*, *CCR5*, *AAVS1* and *CCR2* fragments were amplified with primer sets 4684-YJ170R, YJ207F-YJ208R (or YJ207F-YJ225R), YJ222F-YJ223R and YJ220F-YJ359R, respectively, using the Phusion High-Fidelity PCR Master Mix (Thermo Scientific, Waltham, Massachusetts). After denaturation and re-annealing, each sample was digested by 1 μl Surveyor nuclease plus 1 μl enhancer (Transgenomic). Cleavage products were separated by gel electrophoresis in 1.5% agarose gel and stained by ethidium bromide. Quantification was based on relative band intensities. Indel percentage was determined by the formula 100 × (1−(1−fraction

cleaved)$^{1/2}$), wherein the fraction cleaved is the sum of the cleavage product peaks divided by the sum of the cleavage product and parent peaks.

## Sequence analysis of targeted loci

To analyze gene disruption by sequencing, we PCR-amplified and cloned target fragments allowing for separate sequencing of plasmid preparations from separate bacterial clones. Fragments encompassing the *egfp*, *CCR5* and *AAVS1* target sites were amplified using primer sets YJ180F-YJ181R, YJ237F-YJ238R, and YJ240F-YJ244R, respectively, from genomic DNA of cells transduced with 600 ng p24 in case of LP-ZFNLR(gfp) and with 200 ng p24 in the case of LP-ZFNLR(CCR5) and LP-ZFNLR(AAVS1). The resulting PCR products were digested with *EcoR*I and *BamH*I (for *egfp* and *CCR5*) or *EcoR*I and *Bgl*II (for *AAVS1*), and cloned into *EcoR*I/*BamH*I-digested pUC57. Sequence analysis was performed on individual cloned transformants using primer YJ226F.

## Targeted gene modification and insertion analysis

Cells transduced with 'all-in-one' vectors IDLV-ZFNLR(CCR5)/donor-LS and IDLV-ZFNLR(AAVS1)/donor-LS were harvested 9 days posttransduction. Genomic DNA was purified as described above. PCR primer sets were designed to identify gene targeting as shown in *Figure 3F* (YJ191F-YJ208R for IDLV-ZFNLR(CCR5)/donor-LS and YJ191F-YJ205R for IDLV-ZFNLR(AAVS1)/donor-LS). Fragments of *AAVS1* and *CCR5* loci were amplified by primer pair YJ222F-YJ223R and YJ224F-YJ225R, respectively, as loading controls.

## Cells and culture conditions

HEK293 (human embryonic kidney cells), 293T, and NHDFs (normal human dermal fibroblasts) were cultured in Dulbecco's Modified Eagle's Medium (Lonza, Basel, Switzerland). Culture medium was supplemented with 10% fetal calf serum, 100 U/ml penicillin, 100 µg/ml streptomycin, and 250 µg/ml L-glutamine. Primary human keratinocytes (HKs) were grown in serum-free keratinocyte medium (Gibco BRL-Life Technologies, Carlsbad, California) supplemented with bovine pituitary extract (25 µg/ml) and recombinant epidermal growth factor (0.1–0.2 ng/ml). All cells were cultured at 37°C and 5% (vol/vol) $CO_2$. To determine the activity of LP-incorporated ZFNs, we established a reporter system in HEK293 cells (HEK293-eGFPmut reporter cells) based on the genomic insertion of a *Sleeping Beauty* DNA transposon (carried by the plasmid pT2/CMV-egfp-mut.SV40-neo) harboring an *egfp* cassette gene from which expression was inhibited by stop- and frameshift-mutations flanked by the two ZFN recognition sequences.

## Flow cytometry

To examine LP-ZFNLR(gfp)-induced homologous repair, HEK293-eGFPmut reporter cells were seeded in six-well plates (1 × 10$^5$ cells/well), and on the following day, cells were co-transfected with 1.8 µg pLV/egfp-donor-fw, 100 ng pcDNA3.1-ZFNL(gfp), 100 ng pcDNA3.1-ZFNR(gfp) as a positive control, or only transfected with 1.8 µg pLV/donor-egfp-fw and 6 hr later transduced with LP-ZFNLR(gfp). To measure homologous repair induced by IDLV-ZFNLR(gfp)/donor, HEK293-eGFPmut reporter cells were seeded in six-well plates at a density of 1 × 10$^5$ cells/well and the next day transduced with VSV-G-positive or VSV-G-negative lentiviral vectors. Reporter cells pretreated with 1 µM Bafilomycin A1 (Baf A1) for 30 min before transduction served as an additional negative control. 4 days after transduction, cells were washed with PBS before being harvested and fixed with 4% paraformaldehyde. Data were collected on a FACSCalibur (Becton Dickinson, Franklin Lakes, New Jersey) and analyzed with FlowJo (Tree Star, Ashland, Oregon).

## Western blot analysis of ZFNs/TALENs incorporated into LPs

To analyze the protein contents of engineered LPs, LP-containing supernatants were centrifuged through a 20% (wt/vol in PBS) sucrose cushion. LPs were lysed in the presence of a protease inhibitor. The viral proteins were separated by SDS-polyacrylamide gel electrophoresis and transferred to the PVDF membrane. Membranes were blocked by 5% fat-free milk dissolved in TBS/0.05% Tween-20 for 1 hr and incubated with an HA monoclonal antibody (Covance, Princeton, New Jersey) overnight at 4°C. The membranes were incubated with anti-mouse secondary antibodies (Dako, Glostrup, Denmark) and visualized by enhanced chemiluminescence (ECL) using a HRP substrate (Thermo Scientific). The HA monoclonal antibody was washed away by stripping buffer (Thermo Scientific), and the PVDF membrane was re-used for incubation with a HIV-1 p24 polyclonal antibody (Thermo Scientific) and later peroxidase-conjugated anti-rabbit secondary antibody (Dako).

## Quantitative PCR for determination of MOIs

Quantitative PCR (qPCR) on lentiviral DNA was performed as previously described (*Bak et al., 2013*). Briefly, qPCR reactions were run using Maxima Probe qPCR Master Mix (Fisher Scientific, Waltham, Massachusetts) with primers and probe specific for the WPRE sequence present in the lentiviral vector. Primers and probe specific for the *albumin* gene were used to quantify the cell number in each qPCR reaction assuming an *albumin* gene copy number of two per genome. WPRE and *albumin* DNA copy numbers were determined using standard curves with serially diluted plasmid containing the two targets. For WPRE, the pCCL-PGK-Puro-H1-MCS plasmid was used (*Bak et al., 2011*) and for *albumin* we used pAlbumin kindly provided by Didier Trono (Addgene plasmid #22037). The multiplicity of infection (MOI) was calculated as: 2 × copy numbers of lentiviral DNA/copy numbers of *albumin*. The infectious units/ng p24 was calculated as: number of cells × MOI/ng p24.

## Acknowledgements

The authors are grateful to Jun Komano for providing packaging constructs for production of fused lentiviral polypeptides. Also, the authors would like to thank Adrian Thrasher for providing pMDlg/pRRE-D64V. We thank Lisbeth Dahl Schrøder for technical assistance throughout the study. We are grateful to Thomas J Corydon for providing equipment for Western blot analysis and to Tina Hindkjær for related technical assistance. We thank Helle Christiansen, Karin Stenderup, and Cecilia Rosada for providing primary keratinocytes.

## Additional information

### Funding

| Funder | Grant reference number | Author |
| --- | --- | --- |
| Lundbeck Foundation | R126-2012-12456 | Jacob Giehm Mikkelsen |
| Novo Nordisk Foundation | | Jacob Giehm Mikkelsen |
| Agnes og Poul Friis Fond | | Jacob Giehm Mikkelsen |
| Hørslev Fonden | | Jacob Giehm Mikkelsen |
| Grosserer L F Foghts Fond | | Jacob Giehm Mikkelsen |

The funder had no role in study design, data collection and interpretation, or the decision to submit the work for publication.

### Author contributions

YC, Conception and design, Acquisition of data, Analysis and interpretation of data, Drafting or revising the article; ROB, JGM, Conception and design, Analysis and interpretation of data, Drafting or revising the article

## Additional files

### Supplementary file

• Supplementary file 1. Oligonucleotides used in this study, 'Targeted genome editing by lentiviral protein transduction of zinc-finger and TAL-effector nucleases'.

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
