## [Decision Letter]

Thank you for sending your work entitled “Targeted genome editing by lentiviral protein transduction of zinc-finger and TAL-effector nucleases” for consideration at *eLife*. Your article has been favorably evaluated by a Senior editor, a Reviewing editor and 3 reviewers.

The following individuals responsible for the peer review of your submission have agreed to reveal their identity: Todd Golub, Reviewing editor; Perry Hackett, peer reviewer. The other two reviewers have chosen to remain anonymous.

The Reviewing editor and other reviewers discussed their comments before reaching this decision, and the Reviewing editor has assembled the following comments to help you prepare a revised submission.

The manuscript describe proof-of-concept for the use of non-replicative lentiviruses to transiently deliver both DNA-specific nucleases (zinc fingers and TALENs) and donor sequences for homologous recombination-dependent gene editing. The goal of reducing off-target effects of the ZFNs/TALENs is a laudable one, and one that would significantly advance the field. The manuscript was clearly written, and the conclusions in general justified.

However, there are several important issues that must be addressed before the work is suitable for publication.

1) The authors claim that higher levels of repair compared to co-transfection of plasmids encoding the two ZFNs. What is the number of nuclease molecules packaged into each LP, and how does this correspond to 300 ng of p24? Also, could the higher indel efficiency induced by LP be due to higher levels of protein present at the time when cells were harvested for analysis? Furthermore, it is generally difficult to compare absolute modification efficiencies across different experiments (especially those previously reported) – cell conditions, transfection efficiencies, and even analysis conditions can vary greatly. If the authors really want to make the argument that there is increased efficiency, they should conduct a side-by-side comparison.

2) In order to be maximally informative for readers, the authors should specify more of the experimental parameters surrounding the work. For example, what was the time point at which the ZFN-LP exposed cells were analyzed compared with transiently transfected cells? Similarly, since viruses were used, provide MOIs of the vectors (from qPCR of the genomes) because without this information we have no idea of the numbers of cells that even have a chance at gene editing, nor the efficiency of those cells with the right genomes. As a result, it is hard to compare efficiencies of 1%, 4% and 8% in the different panels.

3) The indel modification rates for the gels shown should be quantified. Similarly, the Surveyor results in Figure 5 should be quantitated since the image is only marginally convincing. How many times was this experiment repeated? It may be important to repeat the experiment and try and quantitate the results in some way to substantiate the argument that this ZFN pair is safer when delivered by LP. This is an important claim in the manuscript.

4) Figure 3 should be redrawn to make the targeting construct (lentiviral RNA genome) clearer. Where are the homology arms and how are they related to the mutant reporter in the host cell genome? What does the reverse-transcribed DNA look like and what is its architecture relative to the target genomic locus? Clarity of this illustration will help readers design their own experiments, and so it is worth the time to make it as clear and informative as possible.

5) Figure 1 and Figure 4: It would benefit the reader to see a control on the western for plasmid-encoded HA-tagged ZFN or TALEN proteins to compare to what is processed by the virus. As it stands, it is difficult to determine whether the bands not labeled as asterisks are really the expected products and that the ones with asterisks are improperly processed.

6) The authors should show how plasmid transfection compares to LP delivery of ZFNs/TALENs to understand whether LP-delivery is generally more efficient than plasmid delivery for multiple reagents, like is shown in Figure 5, or if the CCR5 ZFNs are somehow special in this regard.

---

## [Author Response]

*1) The authors claim that higher levels of repair compared to co-transfection of plasmids encoding the two ZFNs. What is the number of nuclease molecules packaged into each LP, and how does this correspond to 300 ng of p24? Also, could the higher indel efficiency induced by LP be due to higher levels of protein present at the time when cells were harvested for analysis? Furthermore, it is generally difficult to compare absolute modification efficiencies across different experiments (especially those previously reported)* – *cell conditions, transfection efficiencies, and even analysis conditions can vary greatly. If the authors really want to make the argument that there is increased efficiency, they should conduct a side-by-side comparison*.

We acknowledge the fact that it is challenging to directly compare the editing efficiency using methods based on protein transduction and plasmid transfection. Our original intention was to illustrate to the reader that lentiviral delivery of ZFN proteins provides levels of editing that are comparable with levels of editing obtained by transfection of ZFN-encoding plasmids. Based on the fact that plasmid transfection is so efficient in HEK293 and that editing processes thus should not suffer from suboptimal transfection rates, we reasoned that the comparison would be reasonable. Nevertheless, we acknowledge that transfection rates can be further optimized and basically do not intend to conduct a side-by-side comparison with focus on editing efficiency. However, we feel that it is reasonable to retain the transfection data in Figure 3. Instead, we have revised the wording in the text and now state that “LP-ZFNLR (gfp) induced a level of repair that was comparable to that obtained by the efficient co-transfection of plasmid DNA encoding the two ZFNs” (paragraph 1 of Results section entitled “Targeted genome editing by “all-in-one” lentiviral vectors”).

Also, we agree that a comparison of absolute modification efficiencies across different experiments (including those from previous reports) can be problematic. In the Discussion (fifth paragraph) we have rephrased our argumentation and now state: “Although a direct comparison is not possible, we reason that a high rate of repair after protein transduction reflects…”

We agree with the reviewers that the original manuscript suffered from the lack of details on the number of virus particles that are required to obtain efficient levels of repair. As the combined delivery of ZFNs and donor sequence is central to the studies of repair, we have focused our experimentation and derived calculations on providing the exact numbers of infectious particles (IDLVs) that are required to obtain repair. Please see point 2 for further details. In the revised manuscript, we have added details on the estimated number of Gag protein derivatives in matured lentiviral virions. We cannot at present compare the ‘transducability’ of LPs (without DNA) and IDLVs and therefore will not be able to estimate the exact number of ZFNs in LPs. However, as mentioned in the Discussion, we have previously shown that the uptake of LP-delivered heterologous proteins in transduced cells is very limited relative to cells with plasmid-based expression (Cai et al. PMID:24270790). Hence, although LPs transduce cells with very high efficiency they only deliver small amounts of heterologous proteins to each cell. Thus, we find it unlikely that high efficiency of repair using lentiviral protein transduction reflects unusually high levels of protein at the time when the cells were harvested for analysis of gene disruption or repair. More likely, the editing process may benefit from the high local concentration of ZFNs in virus-derived complexes within transduced cells. We have further strengthened the wording in the Discussion.

*2) In order to be maximally informative for readers, the authors should specify more of the experimental parameters surrounding the work. For example, what was the time point at which the ZFN-LP exposed cells were analyzed compared with transiently transfected cells? Similarly, since viruses were used, provide MOIs of the vectors (from qPCR of the genomes) because without this information we have no idea of the numbers of cells that even have a chance at gene editing, nor the efficiency of those cells with the right genomes. As a result, it is hard to compare efficiencies of 1%, 4% and 8% in the different panels*.

We certainly agree with the reviewers on the importance of providing more detailed information about the number of infectious units that were used to obtain gene repair in transduced cells. We carried out qPCR on lentiviral DNA intermediates (total DNA) in transduced HEK293 cells and determined the number of lentiviral DNA copies relative to the copies of the endogenous albumin gene in IDLV-treated cells. Using this strategy, we obtained exact MOIs for IDLVs carrying the donor sequence (IDLV/donor) and ZFN-loaded IDLVs carrying the donor (IDLV-ZFNLR(gfp)/donor) and found that the overall transduction capacity of ZFN-containing IDLVs was reduced about 5-fold relative to the control. Based on these experiments, we have added MOIs (or details on number of infectious particles) in all IDLV-based repair experiments. Our data show that MOIs in the range of 17-34 used throughout the paper facilitate efficient repair. A new paragraph describing details of the analysis has been added to the Results section “Targeted genome editing by “all-in-one” lentiviral vectors”, and in the Materials and methods section.

*For example, what was the time point at which the ZFN-LP exposed cells were analyzed compared with transiently*
*transfected cells?*

All our gene disruption analyses were performed 24 h after LP transduction or after transfection of plasmid unless specifically pointed out (Figure 1). We have specified the time points in the legends of Figures 1, 2, 4 and 5.

*3) The indel modification rates for the gels shown should be quantified. Similarly, the Surveyor results in*
Figure 5
*should be quantitated since the image is only marginally convincing. How many times was this experiment repeated? It may be important to repeat the experiment and try and quantitate the results in some way to substantiate the argument that this ZFN pair is safer when delivered by LP. This is an important claim in the manuscript*.

We have quantified the indel modification rates, as suggested by the reviewers, and introduced the specific rates in the appropriate figures. In general, these modification rates mimic and support the rates that were obtained by sequencing of individual clones.

We agree that the improved specificity of LP-delivered ZFNs is an important claim and that this issue needed further experimentation. Therefore, we repeated the analysis with new LP preps and carried out the initial analysis of *CCR5*-directed disruption under experimental conditions providing almost similar modification rates for LP- and plasmid-directed ZFN delivery (slightly higher efficiency with protein transduction; 8 % vs*.* 7 %). Under such conditions (and in the same cells), we compared the *CCR2* modification rate and found higher off-target activity within this locus after plasmid-directed ZFN production. Hence, under the given circumstances, plasmid transfection caused a *CCR2* modification rate of 3 %, whereas indels within *CCR2* were not detectable after LP transduction. These lend further support to the notion that the use of LPs is supported by a more favorable on-target/off-target ratio. The new data are presented in Figure 5 and additional text describing this experiment is provided (last paragraph of the Results).

*4)*
Figure 3
*should be redrawn to make the targeting construct (lentiviral RNA genome) clearer. Where are the homology arms and how are they related to the mutant reporter in the host cell genome? What does the reverse-transcribed DNA look like and what is its architecture relative to the target genomic locus? Clarity of this illustration will help readers design their own experiments, and so it is worth the time to make it as clear and informative as possible*.

Yes, we are trying to pass on a lot of information in this figure and agree that the original version was a bit unclear on the mentioned aspects. We revised the figure according to the reviewers suggestions and made the following changes: i) the homologous sequences in the donor and target are now highlighted in light yellow; ii) the shown DNA intermediates are now presented in the same orientation, so that the reverse-transcribed DNA intermediate shown in the nucleus mimics the bigger version shown in the cytoplasm; iii) hatched lines have been added to specify the event of homologous recombination between the lentiviral DNA and the target in the genome. For simplicity, we did not add ‘recombination lines’ for the DNA circles but have added a related comment in the legend. We hope that the figure now succeeds in demonstrating the aspects of co-delivering ZFNs and donor sequence and at the same illustrating the basic design of the genomic target sequence.

*5)*
Figure 1
*and*
Figure 4*: It would benefit the reader to see a control on the western for plasmid-encoded HA-tagged ZFN or TALEN proteins to compare to what is processed by the virus. As it stands, it is difficult to determine whether the bands not labeled as asterisks are really the expected products and that the ones with asterisks are improperly processed*.

This is certainly a valid point. To address this issue, we cloned plasmid constructs expressing the appropriate HA-tagged ZFN and TALEN variants. We then repeated the Western blot analysis of protein derived from matured LPs and included protein from plasmid-transfected cells as the control. The data are consistent with the ZFN and TALEN lengths that were already provided in the original manuscript. The new Western blots for ZFN and TALEN are provided in Figure 1—figure supplement 1 (short and long exposures) and Figure 4, respectively. Notably, these data also confirm that full-length TALEN protein is present in LPs despite the fact that the truncated variants are created by internal cleavage. This observation is consistent with the demonstration of TALEN activity by delivery of TALEN-loaded LPs.

*6) The authors should show how plasmid transfection compares to LP delivery of ZFNs/TALENs to understand whether LP-delivery is generally more efficient than plasmid delivery for multiple reagents, like is shown in*
Figure 5*, or if the CCR5 ZFNs are somehow special in this regard*.

As discussed under 1), further studies addressing different parameters like transfection reagents and cell type would be needed to provide a complete comparison of efficiencies obtained with LP transduction and plasmid transfection. Therefore, in this proof-of-principle study, it was not our intention to directly correlate efficiencies obtained using the two approaches, and we have rephrased sections where the comparison was too direct. Nevertheless, studies of specificity allowed us to directly relate on- and off-targeting obtained with LP delivery and plasmid transfection. We carried out such studies using *CCR5*-directed ZFNs, since the presence of the neighboring *CCR2* locus allowed specific detection of NHEJ-directed deletions and detection of mismatches within the *CCR2* locus.